# Preparation and Characterization of Photoluminescent Graphene Quantum Dots from Watermelon Rind Waste for the Detection of Ferric Ions and Cellular Bio-Imaging Applications

**DOI:** 10.3390/nano12040702

**Published:** 2022-02-20

**Authors:** Chatchai Rodwihok, Tran Van Tam, Won Mook Choi, Mayulee Suwannakaew, Sang Woon Woo, Duangmanee Wongratanaphisan, Han S. Kim

**Affiliations:** 1Civil and Environmental Engineering, Konkuk University, 120 Neungdong-ro, Gwangjin-gu, Seoul 05029, Korea; c.rodwihok@hotmail.com (C.R.); mayulee.suwannakaew@hotmail.com (M.S.); sieg8192@gmail.com (S.W.W.); 2School of Chemical Engineering, University of Ulsan, 93 Daehak-ro, Nam-gu, Ulsan 44160, Korea; bumpatran@gmail.com (T.V.T.); wmchoi98@ulsan.ac.kr (W.M.C.); 3Department of Physics and Materials Science, Faculty of Science, Chiang Mai University, Chiang Mai 50200, Thailand; duangmanee.wong@cmu.ac.th

**Keywords:** photoluminescent graphene quantum dots, watermelon rind waste, hydrothermal treatment, Fe3+ detection, cellular bio-imaging

## Abstract

Graphene quantum dots (GQDs) were synthesized using watermelon rind waste as a photoluminescent (PL) agent for ferric ion (Fe${}^{3+}$) detection and in vitro cellular bio-imaging. A green and simple one-pot hydrothermal technique was employed to prepare the GQDs. Their crystalline structures corresponded to the lattice fringe of graphene, possessing amide, hydroxyl, and carboxyl functional groups. The GQDs exhibited a relatively high quantum yield of approximately 37%. Prominent blue emission under UV excitation and highly selective PL quenching for Fe${}^{3+}$ were observed. Furthermore, Fe${}^{3+}$ could be detected at concentrations as low as 0.28 μM (limit of detection), allowing for high sensitivity toward Fe${}^{3+}$ detection in tap and drinking water samples. In the bio-imaging experiment, the GQDs exhibited a low cytotoxicity for the HeLa cells, and they were clearly illuminated at an excitation wavelength of 405 nm. These results can serve as the basis for developing an environment-friendly, simple, and cost-effective approach of using food waste by converting them into photoluminescent nanomaterials for the detection of metal ions in field water samples and biological cellular studies.

## 1. Introduction

The population increase in South Korea has led to massive agricultural production and increases in the daily domestic consumption, which, in turn, has caused the generation of high volumes of food waste. Food waste disposal has thus become an issue of significant concern in the country. As is well-known, food waste contains high amounts of organic components, making its treatment and disposal expensive and energy-intensive [1]. In addition, landfills and incineration may give rise to other environmental issues, including the emission of toxic gases into the atmosphere and contamination of surface waters, groundwater, and soil [2]. Therefore, food waste re-generation and recycling of food is currently receiving considerable attention as a critical, albeit challenging, topic in the field of environmental science and engineering. Ferric ion (Fe${}^{3+}$) is the one of the trace elements readily found in natural waters and is a common heavy metal contaminant in drinking water and tap water [3]. The intake of high levels of Fe${}^{3+}$ results in malfunctions of human organs, disrupting the physical and chemical processes [4]. Therefore, in the interests of public health, Fe${}^{3+}$ detection in water should be rapid and effective. To quantify the Fe${}^{3+}$ content in water, techniques, such as Mössbauer spectroscopy and inductively coupled plasma-atomic emission spectroscopy, have been proposed [5,6,7]. However, these techniques are characterized by certain limitations, including extensive time consumption and the requirement of highly technical skills, resulting in high costs for analytical operations.

In recent years, nanocarbon materials, such as carbon dots (CDs) and graphene quantum dots (GQDs), have garnered more attention. Owing to their excellent photostability, outstanding biocompatibility, high dispersibility in aqueous media, and non-toxicity, these have diverse applications, such as photocatalysis, energy production, gene therapy, drug delivery, and cancer therapy [8,9,10,11,12,13,14,15,16,17]. Additionally, nanocarbons possess strong fluorescence characteristics and can be used as excellent photoluminescence (PL) probes for ion detection and cellular bio-imaging [18,19,20,21,22,23,24]. Nanocarbon materials have also been reported to be derived from food waste, such as pomelo peels, prawn shells, lemon peels, fish scales, bamboo leaves, and peanut shells [25,26,27,28,29,30]. According to the annual report of the Ministry of Environment of Korea, 14,389 tons of food waste are generated every day in South Korea [31]. Watermelon (*Citrullus lanatus*) is extensively consumed in South Korea in the summer season. Watermelon rind as food waste contains high amounts of proteins, cellulose, pectin, citrulline, and carotenoids [32]. These compounds can be useful as nitrogen- and carbon-rich precursors for preparing nitrogen-doped carbon dots (N-CDs). Interestingly, it has been reported that N-CDs can serve as a superior detector for mercury (II) ion by the annihilation of nonradiative charge-carrier recombination through effective energy or charge transfer [33]. Abundant phenolic and hydroxyl groups on the surfaces of the GQDs could react with Fe${}^{3+}$, which facilitates charge transfer and thus restricts exciton recombination [34,35]. Through the interactions between these functional groups on GQD surfaces and Fe${}^{3+}$, the PL intensity can be significantly quenched by Fe${}^{3+}$. Thus, the GQD PL probe can be utilized as an efficient Fe${}^{3+}$ detector allowing for its widespread use in field applications. Furthermore, this strategy allows the food waste to be used as a cost-effective precursor in the synthesis of GQDs.

Herein, an eco-friendly and simple one-pot hydrothermal technique for the re-use of watermelon rind waste to prepare highly photoluminescent GQDs is presented. The surface morphology and chemical composition of GQDs were symmetrically characterized, and their optical and PL properties were studied to access their interfacial potential in Fe${}^{3+}$ detection. The findings of this study are expected to provide a basis for the (i) synthesis of GQDs via a simple technique, (ii) utilization of food waste as an green and cost-effective option for GQD synthesis, and (iii) application of the as-synthesized GQDs as highly sensitive and performance-efficient carbonaceous precursors for metal ion detection in water and excellent biocompatibility in cellular bio-imaging applications.

## 2. Materials and Methods

### 2.1. Preparation and Characterization of GQDs

Watermelon rinds were obtained from a local market in South Korea and washed using Milli-Q water several times prior to being frozen in a refrigerator for 24 h. The frozen watermelon rinds were homogenized using a blender (DB5000A, Thermo Fisher Scientific, Waltham, MA, USA) operated at 240 strokes/min. The rinds (0.2 g) were then transferred to a 100 mL Teflon-lined autoclaving tube, heated at 160 ${}^{\circ }$C for 12 h, and then cooled down to 20 ${}^{\circ }$C. After the hydrothermal process, the obtained solution was filtered through a 0.45 μm filter paper (Whatman, Buckinghamshire, UK) to remove any solid residues. The resulting reddish orange solution was then dialyzed via dialysis tubing (3.5 kD, Spectrum Lab. Inc., New Brunswick, NJ, USA) against Milli-Q water for 24 h to remove any further impurities, and aqueous GQDs were finally obtained as shown in Figure 1. Detailed information for all reagents and instruments used in this study is provided in Appendix A.

### 2.2. Quantum Yield Evaluation of GQDs

The quantum yield (*Q${}_{S}$*) of GQDs was determined as per a report by Xu et al. [33]. Quinine sulfate (Sigma-Aldrich, St. Louis, MO, USA) in a 0.1 M H${}_{2}$SO${}_{4}$ (Sigma-Aldrich) solution was used as a reference fluorophore of the known quantum yield (*Q${}_{R}$* = 0.54 at 365 nm). The *Q${}_{S}$* of GQDs was calculated as follows:(1)$$Q_{S}=Q_{R}\frac{I_{S}A_{R}{n_{S}}^{2}}{I_{R}A_{S}{n_{R}}^{2}},$$
where *I*, *A*, and *n* represent the measured integrated emission intensity, absorption intensity, and refractive index, respectively. The subscripts “*S*” and “*R*” refer to the sample (GQDs) and standard reference (Quinine sulfate), respectively. The amount of GQDs contained in the solution was determined after solidification by freeze-drying. The solution contained 3 mg of GQDs in 10 mL.

### 2.3. Detection of Metal Ions in Aqueous Solution by GQDs

For the detection of Fe${}^{3+}$, a 1 mL aliquot of GQD solution (1 ppm) was prepared and mixed with 1 mL of various concentrations of Fe${}^{3+}$ in 0.1 M Tris-HCl buffer solution (pH 7.0, Sigma-Aldrich). After 5 min, the mixed solution was examined by fluorescence spectroscopy, and the excitation wavelength 365 nm was used. To gain further insight into the selective detection of GQDs, various metal ion solutions (Mn${}^{2+}$, Cu${}^{2+}$, Mg${}^{3+}$, Co${}^{2+}$, Cd${}^{2+}$, Ba${}^{2+}$, Pb${}^{2+}$, Ni${}^{2+}$, and Zn${}^{2+}$) were prepared and examined via a procedure similar to that employed for the detection of Fe${}^{3+}$.

### 2.4. Detection of Fe${}^{3+}$ in Tap and Drinking Water Samples by GQDs

To verify the PL probe performance for the detection of Fe${}^{3+}$ in practical applications (household use), the GQDs were applied to water samples (tap and drinking water samples with no pretreatment, *n* = 3, respectively) collected from a household area. Fe${}^{3+}$ solutions, with varying concentrations of the ion, were added to the water samples and the PL emission spectra were monitored via fluorescence spectroscopy.

### 2.5. Cytotoxicity Test and Cellular Bio-Imaging

The cytotoxicity of GQDs on HeLa cells was evaluated using the methylimidazole tetrazolium (MTT, Sigma-Aldrich) cell viability assay method. The cells were seeded on 96-well plates at 1 × 10${}^{5}$ cells/well in 100 μL of cell medium containing 10% phosphate-buffered saline (PBS, pH 7.4, Sigma-Aldrich). The plates were cultured at 37 ${}^{\circ }$C in a humidified 5% CO${}_{2}$ atmosphere. After 12 h, Dulbecco’s Modified Eagle’s Medium (DMEM, Sigma-Aldrich) was removed, and then cell medium was added, which contained different concentrations of GQDs ranging from 0.5–3 mg/mL. The cells were further incubated for 12 h. Then, 20 μL of MTT solution (5 mg/mL) was added to each well following incubation for 4 h. After removing the culture medium, 150 μL DMSO solution was injected to each well. The untreated HeLa cells were used as the control sample, which was treated under the same conditions. The absorbance of each well was measured at 490 nm. The optical density (OD) of the mixture solution was measured using the Microplate Reader Model (Spectra Max M2, Molecular Devices, CA, USA) three times. The cell viability was calculated using Equation (Equation 2) as follows [36];
(2)$$Cell\;viability\;\left[\%\right]=\frac{OD_{Treated}}{OD_{Control}}\times 100,$$

To investigate the fluorescence imaging of GQDs, HeLa cells at an initial density of 1 × 10${}^{5}$ cells/well were seeded and cultured in the same way as for the cytotoxicity assay process. After the cells were plated, the culture medium was replaced by 2 mL of fresh medium 0.1 mg/mL GQDs, and 30 μM PBS was introduced and incubated for 6 h. To prepare for the confocal imaging experiments, the cells were rinsed three times with the PBS buffer to remove the free GQDs, followed by fixation with 4% paraformaldehyde (Sigma-Aldrich) solution for 15 min and then rinsed twice with the PBS buffer. The fluorescence was observed using a confocal laser scanning microscope at an excitation wavelength of 405 nm (Olympus FV-1200-OSR, Oylmpus, Tokyo, Japan).

## 3. Results and Discussions

### 3.1. Characterization of GQDs

The GODs were formed by a sophisticated process that involves a complex interaction among the resource compositions and internal synthesis environments. During the hydrothermal process, the watermelon rind, as a carbonaceous and nitrogenous source, was heated to a mild temperature. The large carbon and nitrogen cores, such as cellulose, proteins, pectin, citrulline, and carotenoids as starting materials, were disintegrated to small carbon and nitrogen cores, reacted with each other, and then formed into GQDs [37] as illustrated in Figure 1. TEM characterization of GQDs was conducted to investigate the surface morphology and size distribution as shown in Figure 2. The TEM image showed spherical particles (average diameter of 6.46 nm) in Figure 2a and the relative crystal structure with a lattice spacing of approximately 0.22 nm in Figure 2b, which corresponds to the (1120) lattice fringe of graphene [38].

In order to reveal the surface functional groups of GQDs, FT-IR and Raman analyses were performed as exhibited in Figure 3. As seen in Figure 3a, the FT-IR spectrum was observed the broad absorption band at around 3150–3400 cm${}^{-1}$. This indicates that the N-H bonding formed as a result of the incorporation of nitrogen atoms into GQDs (N-H bond) and the hydroxyl group (-OH) [39]. The detected peak at around 1600–1675 cm${}^{-1}$ corresponds to the C=C aromatic and C=O carboxylic groups [40]. The observed Raman spectrum of GQDs (Figure 3b) revealed the typical characteristics of graphene at 1344 cm${}^{-1}$ and 1597 cm${}^{-1}$, corresponding to the D and G bands. The D band was assigned to the structural defect of hexagonal graphitic layers in plan C=C (carboxylic functional group), while the G band was attributed to *sp*${}^{2}$ carbon-type structure [41]. The second order features corresponding to 2D and D+G bands were observed at 2633 cm${}^{-1}$ and 2927 cm${}^{-1}$, respectively. Along with FT-IR and Raman, this indicates that the amide, hydroxyl, and carboxylic functional groups were predominant in GQDs.

The surface chemical composition of GQDs was further identified using XPS (Figure 3c). Strongly identical components were observed at 286.08, 400.08, and 532.08 eV, corresponding to theC 1s, N 1s, and O 1s peaks, respectively. This suggests that the major compositions of GQDs contain carbon, nitrogen, and oxygen and that the constitution of GQDs is free of contaminants. As evident from the high resolution of the deconvoluted C 1s peak, there were four peaks located at 283.12, 284.00, 285.81, and 288.17 eV, corresponding to C=C, C-N, C-O, and C=O, respectively [40]. In addition, the deconvoluted N 1s spectrum showed two peaks, which were ascribed to N-H${}_{2}$ (398.52 eV) and N-C (400.48 eV) [40]. This indicated the the presence of both pyridine and pyrrolic N atoms on the surface of GQDs, which originated from the disintegration of large carbon and nitrogen cores, such as cellulose, proteins, pectin, citrulline, and carotenoids, and the re-formation of N atoms on the GQD surface [37,42]. These results were in good agreement with the FT-IR and Raman results, confirming that the simple one-pot hydrothermal technique could successfully synthesize GQDs in the presence of amide, hydroxyl, and carboxylic bonding groups. The quantum yield of GQDs was determined using quinine sulfate as a standard reference (54% in 0.1 M H${}_{2}$SO${}_{4}$); it exhibited a high quantum yield of approximately 37% as shown in Table 1.

The optical properties of GQDs were further evaluated by UV visible and PL spectroscopy as demonstrated in Figure 4. UV-visible absorption spectra (Figure 4a) showed two prominent absorption peaks at 334 nm and 468 nm, attributed to the π−π${}^{*}$ transition of *sp*${}^{2}$ hybridized nanocarbon and the $n-\pi^{*}$ transition of nitrogen group on GQD surfaces, respectively. These were attributable to the existence of N-H and hydroxyl and carboxylic bonding groups on GQD surfaces. In addition, UV-visible absorption spectra also exhibited a strong PL emission band at 446.5 nm under UV light irradiation at 365 nm.

To further examine the excitation-dependent PL characteristics, the excitation wavelength was varied from 340 to 480 nm as demonstrated in Figure 4b; the maximum PL intensity of GQDs under different excitation wavelengths was clearly observed in the narrow range from 441 nm to 460 nm, and the PL intensity decreased as the excitation wavelength increased.

### 3.2. Detection of Fe${}^{3+}$ in Water Samples

To thoroughly investigate the selective detection of metal ions, the PL intensity at 446.5 nm of GQDs mixed with various metal ions was detected using the excitation wavelength of 365 nm (Figure 5a). Figure 5b shows the fluorescence intensity ratio (*F/F*${}_{0}$) histogram of GQDs at 100 μM of various metal ions. It can be seen that the PL quenching of GQDs with the addition of Fe${}^{3+}$ showed a significant change, while with the addition of other metal ions, only slight PL quenching was observed.

Notably, the PL intensity was decreased to 43% with 100 μM of Fe${}^{3+}$, while slight or no alteration in PL intensity was observed in the presence of other metal ions. This indicates that the GQDs can be applied as an effective Fe${}^{3+}$ detector using the PL quenching technique. Figure 5c shows that the PL characteristic of GQDs was quenched with different Fe${}^{3+}$ concentrations; the intensity decreased with an increase in the Fe${}^{3+}$ concentration. The PL quenching characteristics of GQDs were analyzed using the Stern–Volmer equation as follows [40];
(3)$$\frac{F_{0}}{F}=1+K_{q}\left[C\right],$$
where *F*${}_{0}$ and *F* represent the fluorescence intensities of GQDs in the absence and presence of Fe${}^{3+}$, respectively; *K*${}_{q}$ is the quenching constant of Fe${}^{3+}$; and [C] is the concentration of Fe${}^{3+}$. According to Equation (Equation 3), *K*${}_{q}$ can be evaluated by the conventional extrapolation of the plot between *F*${}_{0}$*/F* and Fe${}^{3+}$ concentration (Figure 5d).

A Stern–Volmer plot at lower Fe${}^{3+}$ concentrations (0–5 μM, *F* = 1.009 + 0.024*C*) with R${}^{2}$ = 0.99 (Figure 5d, inset) shows a more linear characteristic compared to that observed for the full Fe${}^{3+}$ concentration range of 0–100 μM (*F* = 1.039 + 0.013*C*) with R${}^{2}$ = 0.99. Based on the results, the limit of detection (LOD) of Fe${}^{3+}$ using GQDs was calculated to be 0.28 μM, following the IUPAC 3σ criterion [43]. The hydrothermally prepared GQDs from watermelon rind waste in the present study for Fe${}^{3+}$ detection showed a lower LOD when compared with previous reports (Table A1).

The quenching mechanism of Fe${}^{3+}$ was evaluated using UV-visible spectroscopy. The GQDs solution with the presence of Fe${}^{3+}$, the adsorption peak at 334 nm was shifted to 290 nm and the minor adsorption peak at 468 nm disappeared as shown in Figure A1a. In order to further clarify the interaction between GQDs and Fe${}^{3+}$, the resolved fluorescence decay profiles were evaluated by time-correlated single photon counting with a Spex Fluorolog-3 (Figure A1b). The average lifetime ($\tau_{av}$) was calculated using Equation (Equation 4) as follows [44];
(4)$$\tau_{av}=\frac{\sum_{n=1}^{x}\alpha_{n}{\tau_{n}}^{2}}{\sum_{n=1}^{x}\alpha_{n}\tau_{n}},$$
where $\alpha_{n}$ is the percentage of the decay lifetime of $\tau_{n}$ and *n* is the decay time. The fluorescence decay curve was fitted by applying a tri-exponential function. The calculated average lifetime was summarized as in Table A2. It can be seen that the average lifetime of GQDs (6.94 ns) is smaller than that of GQDs with the presence of Fe${}^{3+}$ (5.18 ns). This was attributable to the nonradiative electron of GQDs, which were moved from the excited state to unoccupied orbit of Fe${}^{3+}$ and the effective coordination or chelation between Fe${}^{3+}$ and various amide, hydroxyl, and carboxylic functional groups on the surface of GQDs, and this facilitated the fluorescence quenching [19,45,46].

Compared with the previous researches demonstrating nanocarbon-based PL detection of Fe${}^{3+}$ (Table A1), GQDs derived from waste watermelon rind were found to have superior features in terms of increased sensitivity to Fe${}^{3+}$ (lower detection limit), cost-effectiveness with the utilization of food waste, performance-efficiency, and simplicity of the green and one-pot synthesis technique.

Further applicability of the as-prepared GQDs based on PL detection of Fe${}^{3+}$ was tested using tap and drinking water. GQDs solution was added to the water samples containing various concentration of Fe${}^{3+}$ and the PL emission spectra were observed. The concentrations of Fe${}^{3+}$ in the real water samples were evaluated using the linear equation from Figure 5d. The results are shown in Table 2.

The relative standard deviation (RSD) of the Fe${}^{3+}$ concentration in the real water samples was found to be less than 1%, indicating that the GQDs prepared using waste watermelon rinds can be utilized as a PL probe to detect Fe${}^{3+}$ in the real water samples.

### 3.3. Cytotoxicity Test and Cellular Bioimaging

A standard MTT assay was performed to evaluate the cell viability with different concentrations of GQDs on HeLa cells (Figure 6a). GQDs showed more than 80% cell viability at concentrations up to 3 mg/mL after 24 and 48 h. This result indicates a low toxicity to HeLa cells and excellent biocompatibility of GQDs in bio-imaging applications. To investigate a practical cellular bio-imaging application, GQDs were used as photoluminescence agents in in vitro cellular bio-imaging. Confocal laser scanning microscopy was conducted to image of the HeLa cells with GQD uptake. GQD penetration into the cells through the ion channels and aquaporins of the cell membrane may be influenced by the electric charge/concentration gradient [47].

The HeLa cells were clearly illuminated under an excitation wavelength of 405 nm. When compared with a bright-field micrograph (Figure 6b), GQDs were mostly distributed in the cell nucleus area without accumulating on microtubules and cell membranes (Figure 6c). These results, therefore, suggest that GQDs are a candidate material with excellent biocompatibility for cellular bio-imaging applications.

## 4. Conclusions

A green and simple one-pot hydrothermal technique was used to prepare GQDs with a relatively high quantum yield (c.a. 37%) from watermelon rind waste. Their structures corresponded to the lattice fringe of graphene and possessed various functional groups. GQDs exhibited blue emission under UV excitation and highly selective Fe${}^{3+}$ detection with a low concentration when tested using tap and drinking water samples. GQDs were also successfully applied as a photoluminescence agent in in vitro cellular bio-imaging with a low cytotoxicity. Hence, the GQDs prepared in our study can be used as non-toxic, highly biocompatible, cost-effective, and performance-efficient precursors for metal ion analysis in water and cellular bioimaging applications.

## Figures and Tables

**Figure 1 nanomaterials-12-00702-f001:**
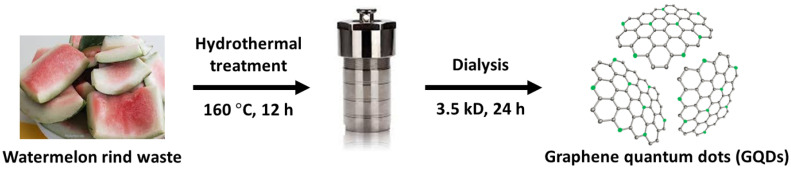
Preparation of GQDs from watermelon rind waste through the hydrothermal technique.

**Figure 2 nanomaterials-12-00702-f002:**
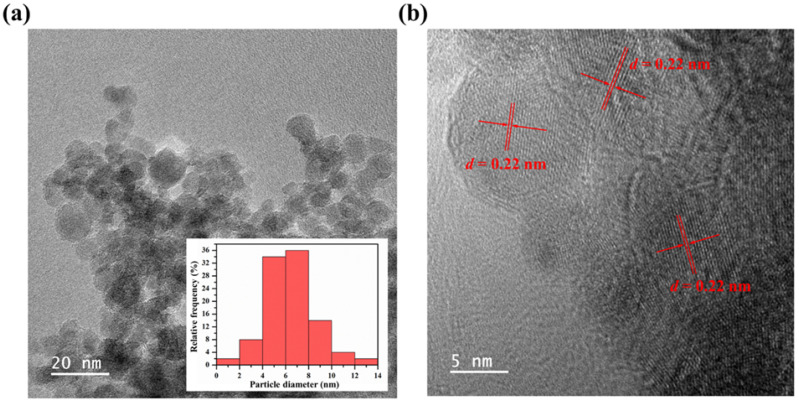
(**a**) TEM image and the size distribution (inset) of GQDs and (**b**) HR-TEM image of GQDs and a lattice spacing of approximately 0.22 nm.

**Figure 3 nanomaterials-12-00702-f003:**
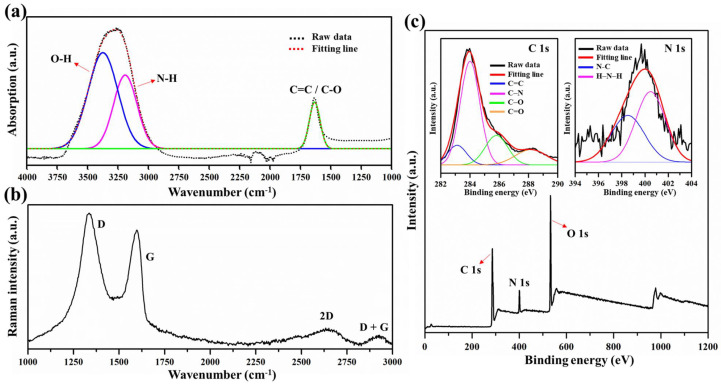
(**a**) FT-IR spectra, (**b**) Raman spectra, and (**c**) broad range XPS spectra of GQDs, the inset: the deconvoluted C 1s and N 1s.

**Figure 4 nanomaterials-12-00702-f004:**
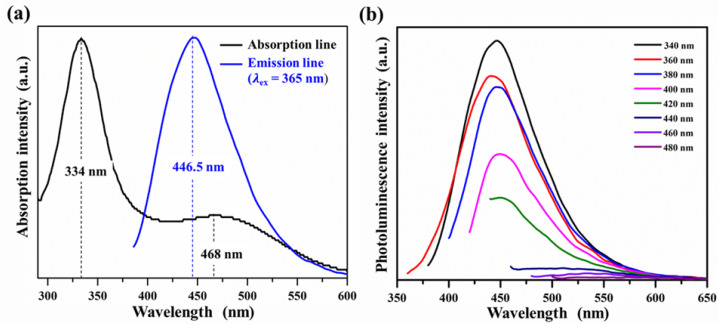
(**a**) UV-vis absorbance spectra of GQDs and (**b**) PL emission spectra of GQDs at different excitation wavelengths.

**Figure 5 nanomaterials-12-00702-f005:**
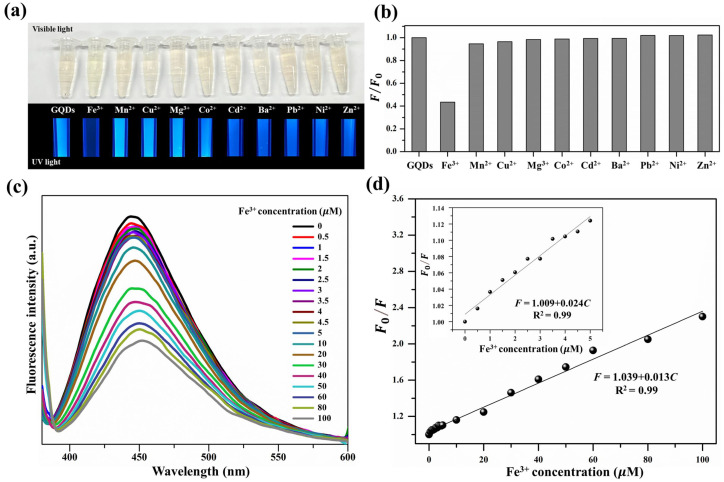
(**a**) PL response of GQDs with various metal ions under an excitation wavelength of 365 nm (top: under visible light; bottom: under UV light), (**b**) fluorescence intensity ratio (*F/F*${}_{0}$) of GQDs at 100 μM of various metal ions, (**c**) PL intensity of GQDs with the addition of Fe${}^{3+}$ in various concentrations, and (**d**) PL intensity of GQDs with different concentrations of Fe${}^{3+}$ (0–100 μM), the inset: the plot of the GQD PL intensity with low concentrations of Fe${}^{3+}$ (0–5 μM).

**Figure 6 nanomaterials-12-00702-f006:**
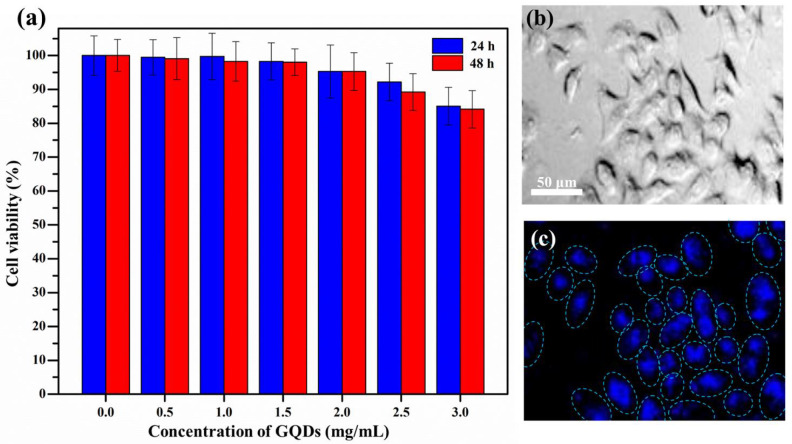
(**a**) Cellular cytotoxicity of (GQDs) towards HeLa cells using an MTT assay and micrographs of HeLa cells under (**b**) bright-field and (**c**) 405 nm of excitation wavelength illumination.

**Table 1 nanomaterials-12-00702-t001:** Quantum yield of GQDs.

Sample	Integrated Emission Intensity	Absorbance at 365 nm	Refractive Index of Solvent	Quantum Yield at 365 nm
Quinine sulfate	30,110 (*I${}_{R}$*)	1.59 (*A${}_{R}$*)	1.33 (*n${}_{R}$*)	0.54 (*Q${}_{R}$)*
GQDs	7593 (*I${}_{S}$*)	0.59 (*A${}_{S}$*)	1.33 (*n${}_{S}$*)	0.37 (*Q${}_{S}$*)

Note: *I*, *A*, and *n* represent the measured integrated emission intensity, absorption intensity, and refractive index, respectively. The subscripts “*S*” and “*R*” refer to the sample (GQDs) and standard reference (Quinine sulfate).

**Table 2 nanomaterials-12-00702-t002:** Fe${}^{3+}$ concentrations and recovery percentage in tap and drinking water samples.

Sample	Standard Fe${}^{3+}$ Concentration (μM)	Detected Fe${}^{3+}$ Concentration (μM)	Recovery Percentage of Fe${}^{3+}$ (%)	Relative Standard Deviation (%)
Tap water	80	77.63	97.04	0.68
		78.51	98.14	
		77.55	96.94	
	50	47.83	95.66	0.98
		48.02	96.04	
		48.72	97.45	
	10	9.55	95.47	1.12
		9.67	96.74	
		9.76	97.63	
Drinking water	80	78.66	97.04	0.68
		77.58	96.98	
		78.03	97.54	
	50	48.93	97.86	0.34
		49.06	98.12	
		48.73	97.46	
	10	9.72	97.23	0.50
		9.74	97.36	
		9.57	95.68	

## Data Availability

Not applicable.

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
