# Peer review of "Preparation and Characterization of Photoluminescent Graphene Quantum Dots from Watermelon Rind Waste for the Detection of Ferric Ions and Cellular Bio-Imaging Applications"

_nanomaterials, 2022, doi:10.3390/nano12040702_

Round 1

Reviewer 1 Report

The authors report the preparation of graphene quantum (GQDs) from watermelon rind waste and their use for (1) the detection of Fe3+ ions in aqueous solution and (2) bioimaging. Some of the results presented may be of interest but the manuscript must markedly be improved before acceptance. Here are my comments :

  • along the whole manuscript, results should be better discussed in the context of literature.
  • GQDs are well known to complex metal ions like Fe3+. Add a reference (Nanomaterials 2020, 10, 104).
  • numerous papers already described Fe3+ detection using GQDs (see for example, Adv. Funct. Mater. 2014, 24, 3021-3026; ACS Appl. Mater. Interfaces 2015, 7, 23958-23966; Anal. Chem. 2014, 86, 10201-10207; ACS Appl. Mater. Interfaces 2016, 8, 21002-21010; ...). The authors must compare the Fe3+ sensing performance of their GQDs to previous reports (concentration of the probes, limits of detection,...) and highlight the advances made. A table should be added.
  • line 60 : "symmetrically" (?)
  • line 89 : how was the GQDs concentration determined ? The text should be clarified.
  • Figure 2 : GQDs are not well dispersed but agglomerated. If possible, better TEM images should be provided.
  • I am surprised by the low number of signals on the FT-IR spectrum. This result is doubtful and should be checked by the authors.
  • line 155 to 158  : the sentence should be moved after the paragraph describing the optical properties of the dots.
  • line 194 : correct adsorption into absorption.

Author Response

Responses to the reviewers’ comments, have been attached here

Reviewer 2 Report

Rodwihok et al. have synthesized, characterized and evaluated the photoluminescent graphene quantum dots from watermelon rind waste for detection of ferric ion in water and application in cellular bioimaging. The work has been carried out methodically and well-written with sufficient scientific inputs. However, the following points need to addressed before this paper could be accepted for publication:

  1. Title should be modified as “Preparation and characterization of photoluminescent graphene quantum dots from watermelon rind waste for detection of ferric ion and cellular bioimaging applications” or simply “Photoluminescent graphene quantum dots from watermelon rind waste for detection of ferric ion and cellular bioimaging applications”.
  2. The authors should provide global and South Korea statistics of watermelon rind waste generation annually.
  3. P2, L60 – What do mean by “symmetrically characterized”?
  4. P3, L81 – Is it “as per Xu’s report” or “as per a report by Xu et al. [28]”.
  5. Section 2.1 – if the synthesis of GQDs is based on a reported method, a reference should be cited.
  6. Section 2.3 and 2.4 – please mention what ferric salt and other metal ion salts used in these studies?
  7. Section 2.3 – this section’s title can be “Detection of metal ions in aqueous solution by GQDs”.
  8. Section 2.4 – this section’s title can be “Detection of Fe3+ in tap and drinking water samples by GQDs”.
  9. P4, L125 – “(Olympus FV-1200-OSR microscope, Olympus Korea, Seoul, South Korea)” should be ““(Olympus FV-1200-OSR microscope, Seoul, South Korea)”.
  10. The authors should address how the PL properties GQDs would be affected by different pH.
  11. Section 3 – the first sub-section under section 3 can be about some details of synthesis such as mechanism description of formation of quantum dots and optimization of conditions as well as some comparison with other synthesis procedures.
  12. Figure 3 & 5 – the inset pictures in Fig. 3c as well as Fig. 5c and 5d are not clear with the labels and they should be enhanced for clarity.
  13. Tables 1, 2 & A2 – The abbreviations in these tables should be described in full form in the footnote of respective tables.
  14. Table 2 – What is the original concentration of iron in tap water and drinking water?

Author Response

(The authors gave the same response as above.)

Reviewer 3 Report

The manuscript entitled, ‘High performance photoluminescent graphene quantum dots from watermelon rind waste for detection of ferric ion and cellular bioimaging applications’ reported synthesis of graphene quantum dots and its sensing and bioimaging applications. I am mentioning some loopholes of this work which should be accounted prior to publication;

  1. The abstract is too informative. It will be better if the author make a concise form of abstract.
  2. The XPS survey showed presence of nitrogen. That should be explained.
  3. It will be better if the author make one comparative table with the other’s works.
  4. The mechanism of capture or sense the ferric ion should be elaborated more. Is there any chance of chelation between ferric ion and GQDs?
  5. Did they check this probe in nM? Is that work? Just discussion. No need to do further experiments.
  6. Several articles related to this could be nurtured for more literature survey. I am mentioning some of them: DOI: 1039/D1NA00447F; https://doi.org/10.1016/j.foodchem.2020.128893; https://doi.org/10.1021/acsabm.0c01104.

Author Response

(The authors gave the same response as above.)

Round 2

Reviewer 1 Report

All corrections were made by the authors. The language can still be improved but these corrections can be made at the proof stage.

The manuscript can be accepted by Nanomaterials.

Reviewer 3 Report

This can be accepted in its present form.